# Changes and Variations in Death Due to Senility in Japan

**DOI:** 10.3390/healthcare8040443

**Published:** 2020-10-30

**Authors:** Masaki Bando, Nobuyuki Miyatake, Hiroaki Kataoka, Hiroshi Kinoshita, Naoko Tanaka, Hiromi Suzuki, Akihiko Katayama

**Affiliations:** 1Department of Hygiene, Faculty of Medicine, Kagawa University, Miki-cho, Kagawa 761-0793, Japan; miyarin@med.kagawa-u.ac.jp (N.M.); tanzuki@med.kagawa-u.ac.jp (H.S.); 2Department of Physical Therapy, Kenshokai College of Health and Welfare, Tokushima 760-0093, Japan; 3Department of Physical Therapy, Faculty of Health Sciences, Okayama Healthcare Professional University, Okayama 700-0913, Japan; h.kataoka59@gmail.com; 4Department of Forensic Medicine, Faculty of Medicine, Kagawa University, Miki-cho, Kagawa 761-0793, Japan; kinochin@med.kagawa-u.ac.jp (H.K.); ntanaka@med.kagawa-u.ac.jp (N.T.); 5Faculty of Social Studies, Shikokugakuin University, Zentsuji City 765-0013, Japan; kata@med.kagawa-u.ac.jp

**Keywords:** senility, Jointpoint analysis, coefficient of variation

## Abstract

Objective: The proportion of elderly individuals (≥65 years old) in Japan has markedly increased. However, the definition of senility in Japan is controversial. The aim of the present study was to investigate changes and variations in the number of deaths due to senility in Japan. Methods: Information on the number of deaths due to senility between 1995 and 2018 as well as other major causes of death was obtained from the Statistics Bureau of Japan official website. Changes and variations in the number of deaths due to senility were compared with other major causes of death in Japan. The relationships between the number of deaths due to senility and socioeconomic factors were also examined in an ecological study. Results: The number of deaths due to senility was 35.7 ± 23.2/one hundred thousand people/year during the observation period and has continued to increase. A change point was identified in 2004 by a Jointpoint regression analysis. Variations in the number of deaths due to senility, which were evaluated by a coefficient of variation, were significantly greater than those due to other major causes of death, i.e., malignant neoplasm, heart diseases, cerebrovascular diseases, and pneumonia. The number of elderly individuals (≥65 years old) (%) and medical bills per elderly subject (≥75 years old) correlated with the number of deaths due to senility. Conclusion: The number of deaths due to senility has been increasing, particularly since 2004. However, variations in the number of deaths due to senility were observed among all prefectures in Japan.

## 1. Introduction

The proportion of elderly individuals (≥65 years old) in the total population of Japan has markedly increased, with 28.1% being older than 65 years in 2018 [1]. Among 1,362,482 deaths recorded in 2018 in Japan, the five major causes of death were malignant neoplasm (27.4%), heart diseases (15.3%), senility (8.0%), cerebrovascular diseases (7.9%), and pneumonia (6.9%) [2]. In turn, although older-age mortality will increase as the population ages, deaths will start to level off as the population shrinks in Japan, which is part of the demographic transition [3]. The number of deaths is expected to increase for some time in Japan [1]. Therefore, appropriate strategies for maintaining the health of the elderly are urgently needed.

Senility was the third major cause of death in Japan in 2018. Senility is only in the cause of natural death in the elderly when there is no other cause of death, and the death due to dementia is also defined apart from senility from the view of forensic medicine in the Ministry of Health, Labor and Welfare, Japan [4]. However, the definition of senility in Japan is controversial [5]. Esaki et al. [6] investigated the cause of death in subjects older than 100 years who died based on pathological anatomy, and identified relevant causes other than senile death in every case. Hawley [7] also reported the lack of senile deaths using a pathological autopsy. Gessert et al. [8] examined causes of death from the death certificates of 26,415 individuals aged 70 years and older, and suggested that death certificates need to be modified to facilitate the direct acknowledgment of age-related frailty as a contributing cause of death. In Japan, death due to senility is diagnosed only in the case of death due to natural causes, where there is no other cause of death in the elderly [4]. A study compared physicians with and without any experience of diagnosing death due to senility and demonstrated that there was no difference between the groups in terms of the average years of experience as a physician and years of experience in providing home care [9]. This suggests that rather than the level of experience, the beliefs and perspectives of the physicians engaged in terminal medical care have an impact on the number of death due to senility. Although it is one of the major causes of death in Japan, limited information is currently available on senility [10,11,12,13], particularly in recent years.

Therefore, in this study, we aimed to investigate the changes in the number of deaths due to senility, and to compare variations in the number of deaths due to senility with other major causes of death, i.e., malignant neoplasm, heart diseases, cerebrovascular diseases, and pneumonia in Japan.

## 2. Methods

### 2.1. Number of Deaths

Information on the number (/one hundred thousand people/year) of deaths due to the 5 major causes, i.e., malignant neoplasm, heart diseases, cerebrovascular diseases, pneumonia, and senility, between 1995 and 2018 (24 years) in 47 prefectures in Japan was obtained from the Statistics Bureau of Japan official website [14] (Table 1, Figure 1).

### 2.2. Socioeconomic Factors

We also obtained information on socioeconomic factors, such as the number of elderly individuals (≥65 years old) (%), the number of single-person households (%), household income (Japanese yen), and medical bills per elderly subject (≥75 years old) (Japanese yen), which are considered to affect senility, from an official Japanese governmental website [15]. All parameters, except for the number of single-person households (%), were reported in 2016. The number of single-person households (%) was reported in 2015, but was not surveyed in 2016.

### 2.3. Ethics

The number of deaths due to major causes and socioeconomic factors were obtained from an official website. This study was approved by the ethical committee of Shikoku Gakuin University, Zentsuji city, Kagawa prefecture, Japan (approval number: 2020002, approval date: 8 September 2020).

### 2.4. Statistical Analysis

The number of deaths due to major causes in Japan (per 100,000 in 47 prefectures in every year) between 1995 and 2018 (24 years) were expressed as a mean ± standard deviation (SD). A trend analysis of the number of deaths due to senility was conducted by the Jointpoint regression program, 4.8.0.1 (National Cancer Institute) [16,17]. To compare variations in the number of deaths due to senility with other major causes of death in Japan, we calculated the coefficient of variation (CV: standard deviation/mean) of 5 major causes of death each year between 1995 and 2018 in Japan, and compared CV (among 47 prefectures for 24 years) using the Kruskal-Wallis test and Steel test [18]. We also examined the relationship between the number of deaths due to senility and socioeconomic factors in Japan using simple and multiple regression analyses, where *p* < 0.05 was significant. Correlation coefficient and partial correlation coefficient were calculated. Statistical analyses were performed using JMP Pro version 15 (SAS Institute Inc., Cary, NC, USA).

## 3. Results

Table 1 and Figure 1 show the number of deaths due to the five major causes between 1995 and 2018 in Japan. The total number of deaths due to malignant neoplasm was the highest among the five major causes, followed by heart diseases, cerebrovascular diseases, pneumonia, and senility. However, the number of deaths due to senility increased in recent years (Figure 1).

We investigated changes in the number of deaths due to senility by a Jointpoint regression analysis, which revealed a change point (2004) that was followed by marked increases (Figure 2).

We then examined variations in the number of deaths due to the five major causes in Japan using CV (Table 2). The CV of senility was significantly higher than those of the other major causes during the time period analyzed (1995–2018). In addition, a change point (2004) was identified by a Jointpoint regression analysis. Therefore, we analyzed two periods (1995–2004 and 2005–2018), and found that the CV of senility was also higher than those of other causes in 1995–2004 and 2005–2018.

To identify the factors affecting the number of deaths due to senility, we examined socioeconomic factors, such as the number of elderly individuals (≥65 years old) (%), the number of single-person households (%), household income (Japanese yen), and medical bills per elderly subject (≥75 years old) (Japanese yen) in Japan, which are considered to be clinically important in 2016 (Table 3) in total, for men and women. A simple regression analysis showed that the number of elderly individuals (≥65 years old) (%), the number of single-person households (%), and medical bills per elderly subject (≥75 years old) (Japanese yen) correlated with the number of deaths due to senility in both sexes. Partial correlation coefficient between the number of deaths due to senility and the number of elderly individuals (≥65 years old) (%) was 0.696 (*p* < 0.001), and between the number of deaths due to senility and the medical bills per elderly subject (≥75 years old) (Japanese yen) was −0.512 (*p* = 0.001). In addition, we further analyzed this by using the number of deaths due to senility in 2017 and 2018. Correlation coefficient between the number of deaths due to senility and the number of elderly individuals (≥65 years old) (%) was 0.642 (*p* < 0.001) in 2017 and 0.657 (*p* < 0.001) in 2018, and correlation coefficient between the number of deaths due to senility and medical bills per elderly subject (≥75 years old) (Japanese yen) was −0.453 (*p* = 0.001) in 2017 and −0.434 (*p* = 0.002) in 2018.

In a multiple regression analysis, we used the number of deaths due to senility as a dependent variable and the number of elderly individuals (≥65 years old) (%), the number of single-person households (%), and medical bills per elderly subject (≥75 years old) (Japanese yen) as independent variables (Table 4). We found that the number of elderly individuals (≥65 years old) (%) and medical bills per elderly subject (≥75 years old) (Japanese yen) were important factors affecting the number of deaths due to senility in both sexes.

## 4. Discussion

In the present study, we examined changes and variations in the number of deaths due to senility in Japan. After 2004, the number of deaths due to senility significantly increased. The CV of the number of deaths due to senility was the highest among the five major causes of death in Japan.

The number of deaths due to senility has decreased since the 1950s [14]. Death due to senility was traditionally pronounced when there was no specific cause of death in the elderly, and was considered to be affected by the level of medical care. After the 1950s, improvements in medical treatment reduced the number of deaths due to senility. In the present study, the Jointpoint regression analysis revealed that the number of deaths due to senility increased after 2004. The number of elderly individuals and their mortality rate have both been increasing in Japan [2]. In the present study, simple and multiple regression analyses identified the number of elderly individuals as an important factor affecting the number of deaths due to senility. Collectively, the present results and previous findings indicate that the number of deaths due to senility will continue to increase in Japan.

As previous studies showed that the deaths due to senility is controversial [5], in the present study, the CV of senility was the highest among the five major causes, indicating variations in the definition of death due to senility. However, the CV of senility in 2005–2018 was lower than that in 1995–2004. The fact that the CV decreases in 2005–2018 may be important, and it shows that there is less variance in the more current data. Progression in diagnosis and better cause of death other than senility might affect these results.

In the present study, simple and multiple regression analyses showed that medical bills per elderly subject was one of the important factors affecting the number of deaths due to senility. Hasegawa et al. [19] reported that the mortality rate in hospitals and number of hospital beds per population were negatively associated with the number of deaths due to senility. Medical resources, including medical bills per elderly subject, the mortality rate in hospitals, and the number of hospital beds per population also appear to be associated with the number of deaths due to senility.

There were a number of potential limitations that need to be addressed. First, this was an ecological study and individual data were not obtained. Second, spurious correlation as anything that relates to an aging population may correlate with old-age mortality. Third, the socioeconomic factors examined in the present study may not have been sufficient. Other factors may affect the number of deaths due to senility. As Japan becomes a much older population, and as public health measures improve, we will see people dying at much older ages. This is a success story. If we cure cancer, heart disease and all other major causes of mortality, there will be an increase in mortality due to senility. Fourth, there was a limited number of years and the danger of extrapolating observations from data from a limited number of years. Validation studies are needed to confirm and extend our findings.

In conclusion, the number of deaths due to senility has increased and variations in senility were noted among the 47 prefectures in aging Japan. The definition of senility needs to be constant and detailed studies by using individual data are urgently required in the future.

## Figures and Tables

**Figure 1 healthcare-08-00443-f001:**
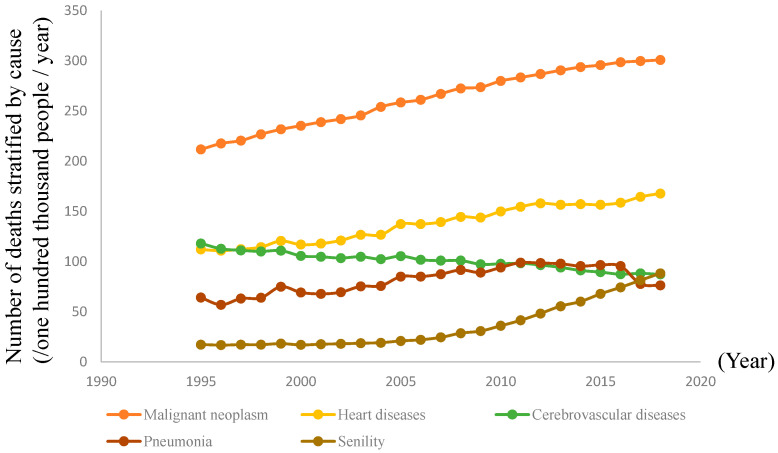
Changes in the number of deaths stratified by cause among all 47 prefectures in Japan.

**Figure 2 healthcare-08-00443-f002:**
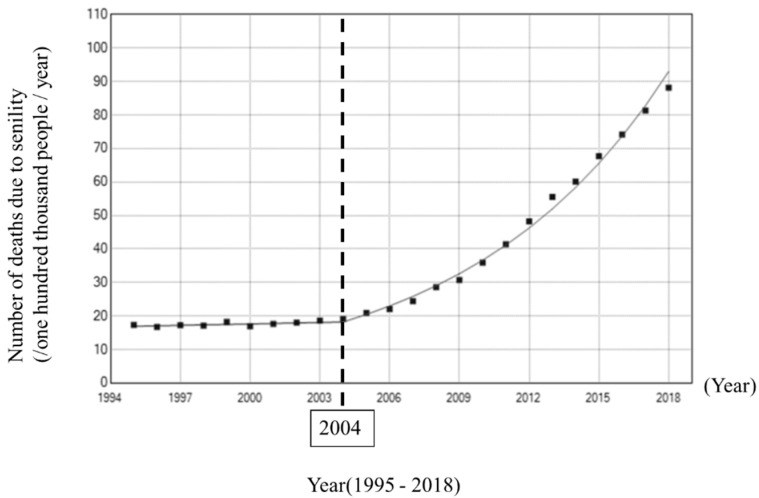
Jointpoint analysis of the number of deaths due to senility.

**Table 1 healthcare-08-00443-t001:** Number of deaths stratified by cause among all 47 prefectures in Japan.

	Mean ± SD			Minimum	Maximum
Number of Years	24				
Malignant neoplasm	261.8	±	28.9	211.6	300.7
Heart diseases	137.6	±	19.0	110.8	167.6
Cerebrovascular diseases	100.8	±	8.5	87.1	117.9
Pneumonia	81.2	±	13.3	56.9	98.9
Senility	35.7	±	23.2	16.7	88.2

Per 100,000 in 47 prefectures in every year for 24 years. SD: standard deviation.

**Table 2 healthcare-08-00443-t002:** Comparison of the coefficient of variation among deaths stratified by cause in all 47 prefectures in Japan.

	All				1994~2004 (Trend 1)				2005~2018 (Trend 2)			
	Mean ± SD			*p*	Mean ± SD			*p*	Mean ± SD			*p*
Malignant neoplasm	0.119	±	0.003	**<0.001**	0.119	±	0.005	**<0.001**	0.119	±	0.002	**<0.001**
Heart diseases	0.162	±	0.012	**<0.001**	0.150	±	0.004	**<0.001**	0.170	±	0.002	**<0.001**
Cerebrovascular diseases	0.219	±	0.011	**<0.001**	0.208	±	0.004	**<0.001**	0.226	±	0.002	**<0.001**
Pneumonia	0.193	±	0.016	**<0.001**	0.182	±	0.009	**<0.001**	0.201	±	0.004	**<0.001**
Senility	0.312	±	0.038		0.349	±	0.019		0.285	±	0.006	

Bold values: *p* < 0.05 vs Senility as control group by the Steel test. SD: standard deviation.

**Table 3 healthcare-08-00443-t003:** Relationships between the number of deaths due to senility and socioeconomic factors.

	All		Men		Women	
	r	*p*	r	*p*	r	*p*
The number of elderly individuals (≧65 years old) (%)	0.652	**<0.001**	0.539	**<0.001**	0.641	**<0.001**
The number of single-person households (%)	−0.460	**0.001**	−0.435	**0.002**	−0.455	**0.001**
Household income (Japanese yen)	−0.016	0.917	−0.001	0.995	−0.003	0.982
Medical bills per elderly subject (≧75 years old) (Japanese yen)	−0.439	**0.002**	−0.518	**<0.001**	−0.439	**0.002**

Bold values: *p* < 0.05 by a simple correlation analysis. r: correlation coefficient.

**Table 4 healthcare-08-00443-t004:** Relationships between the number of deaths due to senility and socioeconomic factors by a multiple regression analysis.

All	B	95% CI		Standardized β	*p*	VIF
Constant	30.662	−34.577	95.901	0.000	0.349	
The number of elderly individuals (≧65 years old) (%)	4.961	3.387	6.534	0.641	**<0.001**	1.169
The number of single-person households (%)	−0.247	−1.475	0.980	−0.047	0.687	1.515
Medical bills per elderly subject (≧75 years old) (Japanese yen)	−0.0001	−0.0001	−4.260	−0.424	**<0.001**	1.352
R^2^ = 0.63, *p* < 0.001						
Men						
Constant	30.540	−9.929	71.009	0.000	0.135	
The number of elderly individuals (≧65 years old) (%)	2.414	1.438	3.390	0.542	**<0.001**	1.169
The number of single-person households (%)	−0.024	−0.785	0.737	−0.008	0.950	1.515
Medical bills per elderly subject (≧75 years old) (Japanese yen)	−0.0001	−0.0001	−0.00003	−0.520	**<0.001**	1.352
R^2^ = 0.56, *p* < 0.001						
Women						
Constant	52.772	−35.718	141.263	0.000	0.236	
The number of elderly individuals (≧65 years old) (%)	6.502	4.367	8.636	0.631	**<0.001**	1.169
The number of single-person households (%)	−0.315	−1.980	1.350	−0.045	0.705	1.515
Medical bills per elderly subject (≧75 years old) (Japanese yen)	−0.0001	−0.0002	−0.0001	−0.425	**<0.001**	1.352
R^2^ = 0.62, *p* < 0.001						

Bold values: *p* < 0.05. 95% CI: 95% Confidence Interval. VIF: Variance Inflation Factor.

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
