# Peer review of "Changes and Variations in Death Due to Senility in Japan"

_healthcare, 2020, doi:10.3390/healthcare8040443_

Round 1

Reviewer 1 Report

Although I found the topic interesting, I found the analyses limiting. A fuller discussion of what senility means is necessary here and to take a very broad view (how useful is it, does it inform policy, does it promote science, does it help identify mortality trends?)

More detailed comments appended:

Line 40. Deaths will start to level off as the population shrinks in Japan. Although older-age mortality will increase as the population ages. This is part of the demographic transition. Edit L 40 to address this observation.

Takahashi, S., Kaneko, R., Ishikawa, A., Ikenoue, M., & Mita, F. (1999). Population projections for Japan: Methods, assumptions and results. Review of Population and Social Policy, 8, 75-115.

L42. The construct of “senility” is undefined in this manuscript. As this is the basis of the paper an elaboration of what is meant by senility, and how it is defined in terms of epidemiological studies is needed.

Traphagan, J. W. (2002). Senility as disintegrated person in Japan. Journal of Cross-Cultural Gerontology, 17(3), 253-267.

I have to read through to L121 to find the definition “Death due to senility was traditionally pronounced when there was no specific cause of death in the elderly, and was considered to be affected by the level of medical care.” Then in L 130 “Esaki et al. [4] investigated the cause of death in subjects older than 100 years who died based on pathological anatomy, and identified relevant causes other than senile death in every case.” Senility is a lazy term in this context, it is used when attending physician is not bothered with defining the real cause of death. You never die of old age, you always die of lack of oxygen to the brain, brought on by other causes (heart disease, cancer etc).

L130 to L141 this section needs to go in the Introduction. This should not be in the Discussion as you already determined your approach.

Mortality from senility is a positive indicator.

L63 “an ecological study” what is meant by this?

L84 Table 1. The table is confusing. Are the figures for “mean”, or “per 100,000” or “mean per 100,000”?

What is “Number of Years 24”? In the second line of the table.

L95 Table 2. What is this table showing? You need to indicate that this is the CV. Is this a Kruskal-Wallis test, with the Steel option making the first group as control? What are you testing for, just proportion of deaths per 100,000? Why are there no probability statistics for the Senility row? Why do the mean (CV?) DECREASE for 2005-2018 rather than increase as you indicated in the narrative Line 93 “CV of senility was also higher”

Senility 0.312 ± 0.038 0.349 ± 0.019 0.285 ± 0.006.

The fact that the SD decreases is also important here as it shows that there is less variance the more current the data. This could be that there is better cause of death other than the generic “old age” which does not mean very much as people cannot die of “old age” (a sociological concept) they have to die of a biological cause.

L105 the analyses are catching broad demographic factors that affect mortality from older age. If you include adult diapers int here you will also find a correlation. This does not inform us about what constitutes senility or more appropriate the aspect of senility we should be looking at. The conclusion where you attempt to bring in the economic issue L150-L154 is also spurious correlation as anything that relates to an aging population will correlate with old-age mortality.

As Japan become a much older population, and as public health measures improve, we will see people dying at much older ages. This is a success story. Saying that there will be higher cause of death due to senility, assumes that this is a negative outcome. If you are going to make this argument then you need to provide a positive alternative (there are no positive alternatives.) If you cure cancer, heart disease and all other major causes of mortality, there will be an increase in mortality due to old age.

Line 141 “In the present study, the CV of senility was the highest among the 5 major causes, indicating variations in the definition of death due to senility.” What does this variation mean? How can you have variations in “cause of death by old age”?

Author Response

To whom this may concern,

Thank you for the thoughtful and constructive feedback you provided regarding our manuscript, Changes and Variations in Death due to Senility in Japan (healthcare- 970163) .

Kind regards

Masaki Bando

Reviewer 2 Report

The article would require some improvements. The introduction is too short so it does not allow to contextualize the topic. It would also require more bibliographic support.

The article does not even cover the breakdown of data by sex and their analysis from a gender perspective. With regard to health in the elderly, there are many studies that support the existence of gender inequalities in health and the need for differentiated analysis.

Author Response

(The authors gave the same response as above.)

Reviewer 3 Report

After a meticulous analysis of the current article, I have found multiple scarcities and errors that could not be accepted in a journal. For this reason, I decided to decline this article.

Next, I will develop all the lacks observed during the analysis:

Abstract:

-First of all abstract should not be structured as objective/results/discussion... it should be a pleasant summary of the article that the reader will read. And in case you decided to structure in this way, you should include an introduction as well.

-Sentences 20-22: you have repeated Japan three times in just two sentences, please delete at least one of them.

-Line 24: This expression "(/one hundred thousand people/year)" is so confused when you use it in the text. In this case, you should write it completely.... "35.7 ± 23.2 hundred thousand people/year" and delete the "/" first of "one hundred..." it looks like the data have to be divided between one hundred, it has no sense.

Introduction:

-The introduction is so poor, you have to context the reader about the topic that you choose, the current state of the situation, and the importance of your research. In this case, any of these points are fulfilled. In addition, just 9 references are so scant, you have consulted just 9 references to decided to research the senility situation in Japan?.

Methods:

-Line 51: you should delete the "/" before "one hundred..." is so confused to the reader and it is not correct.

-Line 60/61: I'm not sure if I understand this paragraph. Do you mean that all the data from 1995 were collected in 2016 or that you only have data for the parameters studied for one year?. In the last case, you cannot do the statistical analysis that you have done. The information for one year is not representative at all, especially on the topic that you have research.

-For last, if you are going to use a Steel test and a Joinpoint analysis, you have to include extra information about the kind of test that they are and which software are you using to carry out them.

Results:

-Table 1: again be careful with the expression "/one hundred..."

-Table 2: I don't understand at all what are you expressing on this table. Because in the previous text I understood that you have calculated the coefficient of variation but you cannot calculate the deviation of the coefficient of variation. On the other hand, you have not explained the Steel test on methods and you have to do it if you are going to use it. For last, it has no sense to calculate p-value here, of course, that you are going to obtain a p-value lower than 0.05, they are the cause of death.

-Table 3: again you are not explained what is "r" or where is coming "p-value". In case that "r" was the correlation coefficient, their values are so poor, which could be related that the mathematical adjusted tested by the authors is not the correct. In addition, you don't talk about the result in the text, in any case, you have to include it. And again, are you using the data for just one year?

-Table 4: a coefficient of regression of 0.63 is really really poor. You cannot accept this result and try to publish it, at least without try other types of mathematical adjusted.

Discussion:

-The discussion is more similar to the introduction that you should include that to the discussion of the results, again you don't include any mathematical data of the result.

Conclusion:

-This section is missing

References:

-Just 14 references for a research article is so scant.

Author Response

(The authors gave the same response as above.)

Round 2

Reviewer 2 Report

The authors have improved the points made in the revision.

In particular, the introduction and results have been extended, so that the purpose of the article is more comprehensible.

Gender analysis continues to be an improvement, as it is not just a matter of separating data by sex, but of introducing the social factors that may be influencing the unequal responses of each gender.

Despite this, the article has been improved and could be considered publication in the journal if the rest of the reviewers consider it.

Author Response

Dear reviewer.

Thank you for the thoughtful and constructive feedback you provided regarding our manuscript, Changes and Variations in Death due to Senility in Japan (healthcare- 970163) .

I am grateful for your support. Thank you.

Reviewer 3 Report

I seriously consider that modifications carried out by the authors are of high interest and facilitate the reading comprehension of the topic of the research. However, the two main reasons because I rejected the article in the first round have not been modified.
First, one of the main points of this investigation is the relationship between the number of death due to senility and socioeconomic factors, but this relation is based on data only from one year. The authors have justified that as a representative year. However, based on my statistical experience, I cannot accept results based on just one year of study, because some important modifications can be observed if more than three years are included, and as a consequence, conclusions would not be conclusive.
On the other hand, the coefficients of regression obtained between the number of deaths and some socioeconomic factors are really poor (0.56 - 0.63). The p-values obtained proved that these factors are significant in the number of death (based on only one year) but the influence of these factors on the number of death is not linear. The authors should test other kinds of mathematical regression.

Even the article quality has been enhanced, these two points are important flaws in the final article. For this reason, I have to reject the article again.

Author Response

(The authors gave the same response as above.)
